# Hepatic Oxidative Stress and Cell Death Influenced by Dietary Lipid Levels in a Fresh Teleost

**DOI:** 10.3390/antiox13070808

**Published:** 2024-07-04

**Authors:** Lingjie He, Yupeng Zhang, Quanquan Cao, Hongying Shan, Jiali Zong, Lin Feng, Weidan Jiang, Pei Wu, Juan Zhao, Haifeng Liu, Jun Jiang

**Affiliations:** 1College of Animal Science and Technology, Sichuan Agricultural University, Chengdu 611130, China; 2022202078@stu.sicau.edu.cn (L.H.); 2021202061@stu.sicau.edu.cn (Y.Z.); qq2312159@sicau.edu.cn (Q.C.); 2022302157@stu.sicau.edu.cn (H.S.); 2022302159@stu.sicau.edu.cn (J.Z.); 2Animal Nutrition Institute, Sichuan Agricultural University, Chengdu 611130, China; fenglin@sicau.edu.cn (L.F.); WDJiang@sicau.edu.cn (W.J.); wupei0911@sicau.edu.cn (P.W.); jjun3@hotmail.com (J.Z.); 3Fish Nutrition and Safety Production University Key Laboratory of Sichuan Province, Sichuan Agricultural University, Ya’an 625014, China

**Keywords:** largemouth bass, dietary lipid, growth performance, oxidative stress, liver ferroptosis

## Abstract

Ferroptosis is a form of regulated cell death characterized by iron-dependent lipid peroxidation, affecting physiological and pathological processes. Fatty liver disease associated with metabolic dysfunction is a common pathological condition in aquaculture. However, the exact role and mechanism of ferroptosis in its pathogenesis and progression remains unclear. In this study, an experiment was conducted using different dietary lipid levels in the feeding of largemouth bass (*Micropterus salmoides*) for 11 weeks. The results revealed that the growth performance and whole-body protein content significantly increased with the elevation of dietary lipid levels up to 12%. The activities of antioxidant enzymes as well as the content of GSH (glutathione) in the liver initially increased but later declined as the lipid levels increased; the contents of MDA (malondialdehyde) and GSSG (oxidized glutathione) demonstrated an opposite trend. Moreover, elevating lipid levels in the diet significantly increased liver Fe^2+^ content, as well as the expressions of TF (Transferrin), TFR (Transferrin receptor), ACSL4 (acyl-CoA synthetase long-chain family member 4), LPCAT3 (lysophosphatidylcholine acyltransferase 3), and LOX12 (Lipoxygenase-12), while decreasing the expressions of GPX4 (glutathione peroxidase 4) and SLC7A11 (Solute carrier family 7 member 11). In conclusion, the optimal lipid level is 12.2%, determined by WG-based linear regression. Excess lipid-level diets can up-regulate the ACSL4/LPCAT3/LOX12 axis, induce hepatic oxidative stress and cell death through a ferroptotic-like program, and decrease growth performance.

## 1. Introduction

Largemouth bass (*Micropterus salmoides*), as a carnivorous freshwater fish, has a key role in aquaculture [1]. It is highly prized by consumers for its delectable flesh, high protein content, and low lipid attributes, making it a species of considerable market value. However, the absence of high-quality, reasonably priced feed has emerged as a major constraint hindering aquaculture development [2]. Lipids play a crucial role in providing energy in fish feed. Fish can produce a range of saturated and monounsaturated fatty acids internally; in contrast, fish need to obtain certain essential polyunsaturated fatty acids, which necessitates their intake from dietary sources [3,4]. Previous research has explored the impact of changing dietary lipid levels on growth performance in largemouth bass. However, the results have shown significant variations, leading to confusion in production practices. Optimal feed lipid levels of 8–18% and crude protein levels of 45–51% have been identified in numerous studies for largemouth bass [5,6,7,8,9,10]. Nevertheless, there has been limited investigation into lipid levels in these studies, with more research focused on exploring the appropriate requirements for protein–lipid interactions. Additionally, discrepancies in these recommended lipid levels may be attributed to variations in largemouth bass body weight, lipid sources used in compound feed formulations, and dietary lipid levels. While prior studies have primarily focused on growth and feed utilization, few have investigated liver health and its influencing mechanisms.

The appropriate level of lipids in the diet is also conducive to enhancing the growth outcomes, health, and flesh quality of fish [11,12,13]. Yet, inadequate or excessive feed lipid levels can generate negative repercussions for both growth performance and overall health. Research has reported the adverse effects of a diet rich in lipids on health, including conditions like metabolic syndrome, liver impairment, and fatty liver disease. For example, excessive lipid levels can reduce the antioxidant capacity of largemouth bass [9], resulting in sustained hyperglycemia and decreased muscle quality of the Senegalese Sole (*Solea senegalensis*) [11] and a reduction in the growth performance of the juvenile spotted knifejaw [13]. Fatty liver disease begins with the accumulation of excess fat in the liver and progresses from mild steatosis to more critical situations, for instance steatohepatitis, cirrhosis, and hepatocellular carcinoma [14,15]. It has been indicated that consuming a high-lipid diet for extended periods of time leads to lipid accumulation and metabolic dysfunction-associated fatty liver [16]. However, the underlying causes and mechanisms of lipid metabolic dysfunction-linked fatty liver disease remain largely unclear. However, there is growing evidence that liver ferroptosis plays a role in the progression of fatty liver disease induced by a high-lipid diet [17,18]. Comparative transcriptomic analysis of fatty liver disease in gibel carp (*Carassius auratus gibelio*), which may involve ferroptosis, suggests potential activation [19], yet its mechanism requires further investigation. The involvement of ferroptosis in the onset and advancement of metabolic dysfunction-associated fatty liver disease remains incompletely understood.

Ferroptosis is a type of cell death that relies on iron and lipid hydroperoxides, recognized as a key initiator of inflammation in nonalcoholic steatohepatitis, thereby accelerating the progression of metabolic dysfunction-associated fatty liver disease [18]. Inhibiting ferroptosis is thus considered a crucial factor in preventing the advancement of hepatocyte death. Oxidative stress and excessive iron levels primarily induce ferroptosis by promoting the generation of intracellular reactive oxygen species (ROS). Under the influence of ACSL4, polyunsaturated fatty acids (PUFAs) undergo acylation to generate unsaturated fatty acids with coenzyme A, forming acyl-CoA (PUFAs-CoA). Subsequently, they interact with LPCAT3, an acyltransferase enzyme linked to phospholipids, to facilitate the formation of esterified compounds (PUFA-PL). Ultimately, these PUFA-PL molecules are directly oxidized by oxygenase, initiating lipid peroxidation and fostering ferroptosis [20,21]. Furthermore, GSH is a reducing substrate for GPX4 activity that reduces lipid peroxides to lipols, thereby protecting cells from oxidative stress damage, and SLC7A11 plays an integral role in maintaining intracellular GSH levels. Therefore, GPX4 and SLC7A11 serve as central regulators in ferroptosis by mitigating the formation of lipid peroxides [22,23]. Given that high-lipid diets disrupt the antioxidant system by inducing lipid metabolism disorders through excessive fatty acid intake, we hypothesize that ferroptosis may contribute to the development of metabolic fatty liver disease induced by high-fat diets. This involvement may be linked to the ACSL4-LPCAT3-mediated signaling pathway. Therefore, this study aims to assess the potential impact of varying lipid concentrations on growth outcomes, liver health effects, and potential mechanisms in largemouth bass. The findings from this study hold significant implications for formulating efficient feed and enhancing the health of cultured largemouth bass.

## 2. Materials and Methods

### 2.1. Experimental Diets

Table 1 outlines the formulations of experimental diets. Fish meal, chicken powder, pork powder, gluten, and fermented soybean meal were served as dietary protein sources. Cassava starch was used as the dietary starch source. Soybean oil and phospholipids were added as dietary lipid sources. Six isonitrogenous practical diets with graded levels of lipids (6%, 8%, 12%, 14%, 16%, and 18%, respectively) were formulated. The coarse ingredients were thoroughly pulverized using a vertical ultrafine feed machine (SWFL Series Vertical Shaft Pulverizer, Chengdu, China) and sifted through a sieve with a mesh size of 80. The components were uniformly mixed using a twin-shaft paddle batch mixer until thoroughly blended (SLHSJ05, Nanjing, China). Subsequently, each diet underwent extrusion using a twin-screw extruder (MY-165, Beijing, China) equipped with a 2 mm die. The processing parameters included a screw speed set at 100 rpm, a temperature of 127 °C, and a pressure ranging from 30 to 45 atm. The meals were allowed to air dry before being kept in plastic bags at −20 °C until needed.

### 2.2. Management and Feeding Practices for the Fish

The feeding experiment occurred at the Ya’an Experiment Station, affiliated with Sichuan Agricultural University. Fish were sourced from Sichuan Meishan Weiji Aquatic Seed Technology Co., Ltd. (Meishan, China) and acclimatized in tanks for four weeks. During this period, largemouth bass were fed a commercially available diet. A total of 900 fish with an average initial weight of 14.36 ± 0.10 g were randomly distributed into 18 concrete tanks (2 m × 1 m × 1.5 m), resulting in 50 fish per tank. The fish were fed their designated diets until satiation twice daily (8:00 am and 6:00 pm) for 11 weeks. Additionally, any remaining feed was collected, dried, and weighed to determine feed intake. Each tank was filled with fresh, dechlorinated water at a temperature between 23 and 27 °C and maintained with dissolved oxygen concentrations in the range of 6 to 6.5 mg/L and a pH of 7.0 to 7.5. The Animal Protection Advisory Committee of Sichuan Agricultural University approved the experimental protocol, license number DKY-2018202027.

### 2.3. Sample Collection

At the end of the feeding trial, a 50 mg/L benzocaine solution was applied to anesthetize the fish after a 24 h fasting period. Each group of 9 fish was anesthetized and frozen at −20 °C for the determination of the initial crude body composition, including protein, lipid, ash, and moisture content. Following measurements of length and weight, nine fish from each group had blood drawn using 1 mL syringes from their tail veins. After the drawn blood was placed in a test tube containing an anticoagulant, it was centrifuged at 3000 r for 10 min to extract plasma. The plasma was then frozen at −20 °C in preparation for further biochemical analysis. After the nine fish were dissected, the liver was extracted and precisely weighed. Fresh fish livers were immediately frozen with liquid nitrogen and stored at −80 °C for ensuing biochemical analysis, as well as quantitative Real-time PCR and Western blot analyses. Additionally, 9 fish from each treatment group were chosen and dissected to acquire fresh liver tissue for histological examination.

### 2.4. Biochemical Analysis

Using normal AOAC procedures, the diet moisture, crude lipid, crude protein, and ash contents were measured (AOAC, 2005). Each sample was put in a constant temperature oven at 105 °C for one day to determine the moisture content. The Kjeldahl and Soxhlet methods were used to determine crude protein and crude fat, respectively, and the ash content was determined by the Muffle furnace. Liver tissue samples were homogenized with an 86% ice-cold physiological saline solution to prepare for biochemical assays. After that, the homogenate was centrifuged for 10 min at 2500 r at 4 °C. The supernatant was then gathered and kept for use in additional biochemical tests at −80 °C. Utilizing commercial kits from the Jiancheng Bioengineering Institute, the following parameters were measured: the levels of triglyceride (TG), total cholesterol (TC), malondialdehyde (MDA), protein carbonyl (PC), oxidized glutathione (GSSG), reduced glutathione (GSH), and the activities of Aspartate transaminase (AST), Alanine aminotransferase (ALT), total superoxide dismutase (T-SOD), anti-superoxide anion (ASA), anti-hydroxyl radical (AHR), catalase (CAT), glutathione peroxidase (GPX), glutathione reductase (GR), and glutathione-S-transferase (GST) (Nanjing, China).

### 2.5. Quantitative Real-Time Reverse-Transcriptase Polymerase Chain Reaction (qRT-PCR)

The liver total RNA was extracted using RNAiso Plus kit (Takara, Dalian, China), followed by an assessment of RNA integrity via agarose gel (1%) electrophoresis and spectrophotometric analysis for quality evaluation. Subsequently, the RNA was reverse-transcribed into cDNA using a PrimeScript™ RT reagent Kit (Takara, Shiga, Japan). The sequences of all primers used in this study are listed in Table 2. The Real-time PCR reaction (10 μL) included 5 μL of ChamQ Universal SYBR qPCR Master Mix (Vazyme, Nanjing, China), 3 μL of ddH_2_O, 0.5 μL of each primer, and 1 μL of cDNA. The qRT-PCR protocol in this study was conducted as follows: 95 °C for 2 min, followed by 40 cycles of denaturation 95 °C for 5 s, annealing at the melting temperature (Tm) for 30 s, then 95 °C for 5 s, extension at 65 °C for 5 s, and a final extension at 95 °C for 15 s. The qRT-PCR was performed using a CFX96 Real-Time PCR Detection System (Bio-Rad, Hercules, CA, USA). The primers of largemouth bass are shown in Table 2. β-actin and 18S rRNA were chosen as internal reference genes for normalization. Using the 2^−ΔΔCT^ technique, the results were computed once it was verified that the primers showed an amplification efficiency of about 100% [24].

### 2.6. Protein Extraction and WB Analysis

The liver tissue was homogenized in an ice-cold RIPA Lysis Buffer with PMSF and a protease inhibitor (Beyotime, Shanghai, China). The protein concentration was determined by a BCA protein detection kit (Beyotime, Shanghai, China). The protein samples were separated on an SDS/PAGE gel and subsequently transferred to a polyvinylidene fluoride (PVDF) membrane. After 5% skim milk powder was blocked for 1.5 h, it was incubated at 4 °C overnight with specific primary antibodies. The primary antibodies were as follows: Anti-β-actin (ZEN BIO, Chengdu, China, 380624, 1:2000), Anti-GPX4 (ZEN BIO, 381958, 1:1000), Anti-SLC7A11 (ZEN BIO, 382036, 1:1000), Anti-Transferrin (ABclonal, Chengdu, China, A1448, 1:1000), Anti-Transferrin Receptor (ABclonal, A21622, 1:1000), Anti-ACSL4 (ZEN BIO, R24265, 1:1000), Anti-LOX12 (ABclonal, Chengdu, China, A14703, 1:1000), and Anti-LPCAT3 (ABclonal, Chengdu, China, A17604, 1:2000). Then, the membranes were washed with PBST five times for five minutes each time and incubated with an Anti-rabbit IgG HPR-linked secondary antibody (Cell Signaling Technology, Beijig, China, #7074, 1:2000) for 1 h, followed by washing the membrane with PBST again five times, and then chemiluminescence detection was conducted using the e-BLOT Touch Imager^TM^ System (e-BLOT, Shanghai, China). The gray value was measured by Image J 1.52q software.

### 2.7. Fe^2+^ Content Assay

Liver tissue samples were added to a nine-times homogenate medium, centrifuged at 10,000× *g* for 10 min, and the supernatant was obtained for measurement. The Fe^2+^ concentration in liver tissues was evaluated utilizing a Biochemical Assay Kit (Elabscience, Wuhan, China, E-BC-K773-M) according to the product manual. Ultimately, Fe^2+^ in the liver was computed using the standard curve of the OD value, which was obtained at 593 nm in a microcoder.

### 2.8. Liver Histology

The liver tissue was removed and placed on a white disk using a camera. The tissues were then dehydrated using a succession of ethanol solutions of increasing concentrations, embedded in paraffin, and fixed in 4% paraformaldehyde for a week. Tissue samples were sectioned (4 μm thick), dewaxed, rehydrated, stained, and then closed with a neutral gel. These sections were subjected to microscope analysis.

### 2.9. Statistical Analysis

The data’s normal distribution was confirmed. Data recorded in Excel 2019 were analyzed using variance in SPSS 25.0 (IBM, Chicago, IL, USA). The results were reported as means ± the standard error (SD). Differences among multiple group means were evaluated using Duncan’s multiple-range test, and *p* < 0.05 indicated statistical significance. Guided by the coefficient of determination (R^2^) value, a broken-line model is implemented to estimate the correlation between dietary lipid levels and weight gain (WG), aiming to calculate the optimal dietary lipid level.

## 3. Results

### 3.1. Fish Growth Performance

Table 3 displays the growth performance and feed utilization of fish when they are fed diets containing changing levels of lipids. No differences were observed in initial survival rate (SR), body weight (IBW), PER, and CF among all the treatments (*p* > 0.05). The 12% group exhibited the highest final body weight (FBW), weight gain (WG), specific growth rate (SGR), feed intake (FI), and feed efficiency (FE) values with rising dietary lipid levels (*p* < 0.05). Conversely, the feed conversion ratio (FCR) reached the lowest value (*p* < 0.05). A broken-line analysis revealed that the most effective lipid supplementation level, as determined by weight gain (WG), was 12.2% for largemouth bass (Y = 11.575X + 407.95, R^2^ = 0.9969; Y = −17.765X + 766.25, R^2^ = 0.9129) (Figure 1). Additionally, the levels of the visceral body index (VSI) and hepatosomatic index (HSI) exhibited a gradual increase in tandem with the incremental supplementation of lipids and reached a maximum at 18% (*p* < 0.05).

### 3.2. Fish Proximate Composition

Table 4 shows the whole-body composition of fish as influenced by diets containing different levels of lipids. The lipid treatment had no effect on whole-body moisture content (*p* > 0.05). The whole-body protein contents exhibited an initial improvement, followed by a subsequent decline, while the whole-body lipid content increased significantly and ash content decreased gradually in response to increasing levels of lipids (*p* < 0.05).

### 3.3. Plasma and Liver Parameters

The biochemical parameters of largemouth bass plasma and liver in different lipid-supplemented diets were measured (Table 5). The activity of AST and ALT was associated with the increase in dietary lipid content (*p* < 0.05). When dietary lipid levels rose, the liver’s contents of TG and TC increased considerably, reaching a peak in diets supplied to fish that included 18% lipids (*p* < 0.05).

### 3.4. Liver Examination and Histology

The gross examination and histological appearances of the liver in largemouth bass are depicted in Figure 2. When the lipid content in the diet was below 14%, the liver of the largemouth bass appeared reddish-brown, while in diets with lipid contents of 14%, 16%, and 18%, localized whitening of the liver in largemouth bass was observed. In addition, when fish were fed with a diet containing less than 14% lipid levels, the shape and structure of hepatocytes appeared regular, with spherical nuclei containing nucleoli positioned predominantly in the center of the cells, exhibiting a relatively tight arrangement. In the dietary 14% and 16% lipid levels, the nuclei were observed to have moved towards the outer edge of the hepatocytes, along with instances of cell membrane lysis and a few enlarged cells. At the dietary 18% lipid level, a significant increase in cell enlargement and vacuolar degeneration cells was observed.

### 3.5. Liver Antioxidant Indicators

Substantial variances were noted in hepatic antioxidant indices among the fish consuming diets with varying lipid levels (Table 6). However, MDA and PC contents initially declined and subsequently increased with the increase in lipid level, and the contents in 12% and 14% lipid levels were lower than those in the other groups, respectively (*p* < 0.05). When fish were fed a diet containing 14% lipid levels, their livers showed considerably higher T-SOD activity than when fish were fed diets containing 16% and 18% lipid levels (*p* < 0.05). However, there were no notable variances detected in the activity of T-SOD among the remaining groups (*p* > 0.05). Fish fed with diets containing 12% lipid levels exhibited higher levels of liver CAT activity than those fed with others. The highest ASA and AHR contents in the liver were recorded in the fish fed diets with 14% lipid levels, and they were significantly higher than the other groups (*p* < 0.05). Furthermore, the relevant indexes of GSH-GSSG oxidation balance are shown in Figure 3. The GSH content, GPX4, and SLC7A11 mRNA and protein levels initially increased and then decreased with the rise in lipid levels in the diet, while the variation in GSSG content followed the opposite trend. Additionally, the activities of GPX, GR, and GST exhibited a similar trend to GSH content, reaching their maximum values at 14% and 12% dietary lipid levels.

### 3.6. Liver Fe^2+^ Concentration and the Expressions of TF and TFR

As depicted in Figure 4, the hepatic Fe^2+^ content in largemouth bass gradually increased with the elevation of dietary lipid levels. Moreover, in the group fed with a diet containing 18% lipid levels, the Fe^2+^ content reached its peak, significantly surpassing those in the 6–14% range (*p* < 0.05). Concurrently, the mRNA expressions of TF and TFR were significantly increased with the increase in lipid addition to the diet. Notably, the groups with 14% to 18% lipid content exhibited significantly higher TF mRNA expression compared to the remaining groups (*p* < 0.05), while the groups with 12% to 18% lipid contents demonstrated significantly higher TFR mRNA expression than the other groups (*p* < 0.05). Additionally, similar changes in protein levels were observed.

### 3.7. ACSL4/LPCAT3/LOX12 Axis

The different dietary lipid levels had a significant impact on the mRNA expressions and protein levels of the LOX12, LPCAT3, and ACSL4 axis in the liver of largemouth bass, as Figure 5 shows. The largemouth bass fed diets with 18% lipid levels had the highest mRNA expression of ACSL4, and this expression increased from 6% to 18% dietary lipid levels. Conversely, the mRNA expressions of LPCAT3 and LOX12 exhibited no notable variances among the groups consuming dietary lipid levels ranging from 6% to 18% (*p* > 0.05). However, consumption of 18% dietary lipid levels resulted in a significant upregulation of LPCAT3 mRNA expression compared to the 6% dietary lipid level group (*p* < 0.05). Similarly, 18% lipid levels led to a significant upregulation of LOX12 mRNA expression compared to the groups consuming 6% to 14% dietary lipid levels (*p* < 0.05). The protein levels and mRNA expressions of ACSL4, LPCAT3, and LOX12 exhibited a consistent trend with the varying lipid levels, peaking in the group with 18% lipid content. Similarly, the protein level peaked in the 14% lipid supplemental group, exhibiting a significant elevation compared to that seen in the 6%, 8%, 16%, and 18% lipid supplemental groups (*p* < 0.05).

## 4. Discussion

### 4.1. Effects of Lipid Levels on Growth Performance, Feed Utilization, and Body Composition of Largemouth Bass

As a carnivorous fish, the largemouth bass exhibits a low utilization rate of carbohydrates, necessitating more stringent requirements for alternative energy sources [25]. Lipids serve diverse physiological functions in the metabolism of fish and shrimp, constituting energy substances for their sustenance. In the current study (with 47% crude protein), the highest values for final body weight (FBW), weight gain (WG), specific growth rate (SGR), feed efficiency (FE), feed intake (FI), and lowest feed conversion rate (FCR) were observed in the fish fed the diet containing 12% lipid levels. Furthermore, the dietary group with 12.2% lipid levels (including 6.2% soybean oil) showed a noteworthy improvement in the growth performance of largemouth bass, according to the WG broken-line regression analysis. This aligns with reported crude lipid requirements of 12% [5] and 11.5–14% [8]. Guo et al. (2019) [10] recommended 18.42%, Li et al. (2020) determined it to be 10% [6], and Chen et al. (2023) found it to be 16% [7]. These disparate outcomes could stem from variations in fish size, protein concentrations within compound feed formulations, the lipid sources employed, and the levels of dietary lipids. Similar findings have been reported for several other species as well, such as 12.82% for seabream (*Acanthopagrus schlegelii*) [26], 12.69% for the Japanese seabass (*Lateolabrax japonicas*) [27], and 10.46% to 12.83% for the juvenile spotted knifejaw [13]. A low-lipid diet dramatically hindered the growth performance of largemouth bass, perhaps because it was deficient in important nutrients including fatty acids and fat-soluble vitamins [28]. Conversely, a high level of dietary lipid can hinder the growth performance of largemouth bass, possibly due to factors such as decreased secretion of growth hormones [29], reduced appetite [30], and metabolic dysregulation [31]. Similar findings were observed in other fish species, where excessive dietary lipid levels resulted in diminished growth performance [32].

Additionally, fish subjected to diets containing low and excessive lipid levels exhibited notably diminished food intake compared to other dietary regimens, likely due to the inherent tendency of fish to shun imbalanced or low-energy diets [33]. According to the study’s findings, FE significantly improved when dietary fat levels increased. However, the PER was leveled off with dietary lipid concentration, and the results were in line with earlier research on other fish [34,35,36], indicating that various lipid diets do not enhance the utilization of protein for the synthesis of body proteins in largemouth bass. On the other hand, several species, such as the juvenile spotted knifejaw [13] and the red spotted grouper (*Epinephelus akaara*) [37], showed a protein-sparing effect when fed different lipid diets. These varying results could be attributed to differences in the species and weight of the fish, the lipid sources used in compound feed formulations, and the dietary levels of lipids.

The crude lipid content of fish is a crucial parameter that impacts both the quality and nutritional value of fish, as well as serving as a pivotal determinant for the processing performance of aquatic products and the excessive accumulation of lipids in the visceral cavity, liver, and muscle tissues of fish [38]. Excessive lipid accumulation in fish has the potential to impact processing yields, product quality, and storage stability, thereby affecting its commercial value [11]. In this experiment, an elevated dietary lipid level led to a significant increase in the body lipid content of fish, while whole fish protein contents of dietary 12% and 14% lipid groups were significantly higher than other groups. These results align closely with those observed in grass carp [13], juvenile cobia [34], and the Senegalese Sole [39]. Furthermore, a negative correlation was observed between whole-body lipid and ash contents, while no significant difference was detected in the moisture content among fish with varying lipid levels. Additionally, the 12% dietary lipid level exhibited the highest proportion of whole-body protein, while the excessive lipid diet decreased the protein content of the fish’s body, in line with findings from prior research [10].

### 4.2. High Lipid Level Diet Can Up-Regulate the ACSL4/LPCAT3/LOX12 Axis and Promote Hepatic Oxidative Stress and Ferroptosis

Excessive lipid intake can lead to cellular swelling, nuclear translocation, the accumulation of lipid droplets, and enlarged hepatocytes with increased cell diameter [40]. These effects have been widely documented in various fish species [41], including those observed in the current investigation. Hepatic histopathological analysis in cases where dietary lipid levels surpassed 14% exhibited evident indications of liver injury, characterized by irregularly shaped cells, pronounced lipid vacuole formation, and noticeable hepatic nuclear polarization. Additionally, TC and TG contents in the liver were positively correlated with dietary lipid intake. These suggest that excessive consumption of dietary lipids may impair the liver’s ability to release lipids into the bloodstream and impede both lipoprotein secretion and fatty acid oxidation, thereby initiating a detrimental cycle [26]. The serum biochemical biomarkers of liver injury that are most sensitive are AST and ALT [42]. Assessing the levels of these markers directly contributes to the evaluation of liver health and function. The current investigation found that as dietary fat levels increased, so did the activities of AST and ALT in the serum, and these were significantly elevated when the lipid content exceeded 14%, implying that an overabundance of dietary lipid consumption in fish may result in liver damage. Meanwhile, the HSI is considered a crucial index for fish production, as elevated visceral fat levels may diminish commercial value [32,43]. Notably, it has been observed that the HSI is positively correlated with dietary lipid levels. These results imply that high dietary lipid consumption in fish may cause ectopic lipid accumulation, which in turn may cause liver injury.

Oxidative stress and subsequent lipid peroxidation are key factors in the pathogenesis of nutritional fatty liver disease [44,45,46]. The degree of intracellular oxidative stress can be determined by alterations in MDA, AHR, and ASA levels as well as the activity of antioxidant enzymes (SOD, CAT, GPX, GSH, GR, and GST) [47,48]. GSH is a reducing substrate of GPX4 activity that can reduce lipid peroxides to lipols, thereby protecting cells from oxidative stress damage. SLC7A11 plays an indispensable role in maintaining the intracellular GSH level. GSH depletion may be the primary cause of hepatic iron drooping, which relies on inhibiting GSH synthesis by suppressing the expression of SLC7A11 [49]. According to earlier research, optimal lipid levels in the diet can enhance antioxidant capacity. Experiments conducted both in vivo and in vitro have shown that the activation of oxidative stress signals results in elevated MDA levels and reduced activity of antioxidant enzymes like SOD and GPX [10,50,51]. This imbalance results in excessive ROS production in the body, ultimately leading to liver cell death [52]. Excessive ROS in the body can lead to lipid peroxidation, where MDA, a product of lipid oxidation, serves as a critical marker to assess the extent of lipid peroxidation and the resulting cellular damage [53]. As anticipated, compared to the group with 12% lipid levels, the high-lipid group (with 18% lipid levels) significantly increased MDA and GSSG contents, decreased the activities of T-SOD, GST, CAT, GPX, and GSH content, while inhibiting GPX4 and SLC7A11 mRNA level and protein expressions. These results clarify that fish fed with the ideal dietary lipid level may improve their antioxidant capacity. Conversely, an abundance of dietary lipid intake may prompt oxidative stress, a phenomenon confirmed in black sea bream (*Acanthopagrus schlegelii*) [54].

Ferroptosis is characterized as a type of programmed cell demise marked by the accumulation of lipid peroxides, setting it apart from apoptosis and necrosis [22]. The important role ferroptosis plays in the emergence of metabolic liver disorders has been demonstrated by earlier research, primarily associated with lipid peroxidation reactions triggered by two factors: iron overload and oxidative stress [55,56,57]. Iron plays a crucial physiological role, participating in various cellular processes such as oxygen storage and transport, mitochondrial respiration, DNA replication, and intercellular signaling [58]. In recent years, numerous studies have unearthed a profound correlation between an imbalance in hepatic iron homeostasis and metabolic diseases, particularly lipid metabolism disorders [59,60]. Elevated lipid levels can significantly induce the accumulation of lipids in hepatocytes, subsequently compromising liver function and disrupting the capacity of TF and TFR to efficiently store and transport iron. Consequently, this can result in the deposition of iron into the liver, leading to the exhibition of symptoms of iron overload. In rats fed a high-fat diet (HFD) and cells treated with fatty acids, the liver exhibits an augmented accumulation of iron, characterized by the upregulation of TFR-1 and its reliance on the Irp1 pathway [61]. In our study, we found that when the fat level in the diet is less than the dietary level of 14%, there was no significant change in Fe^2+^ concentration, followed by a significant increase. In other words, high-lipid induction significantly enhances iron accumulation. The nonheme iron, primarily existing in the insoluble Fe^3+^ form, requires reduction to Fe^2+^ for effective absorption. Once bound to TF, Fe^3+^ is recognized by the TFR situated on the cell membrane. Following absorption, iron reduction takes place within the endosome driven by metalloreductase. Subsequently, the DMT1-mediated release of Fe^2+^ from the endosome into the cytoplasm leads to an elevation in the level of free iron [62]. The abundant Fe^2+^ within cells fuels the Fenton reaction and enzymatic oxidation, facilitating the transformation of PUFAs into lipid peroxides, which ultimately triggers cell iron death and further aggravates fatty liver conditions [63]. It was revealed that the expression of the heat shock protein β-1 (HSPB1) can suppress the expression level of TFR, resulting in reduced iron intake and accumulation, thereby decreasing the sensitivity to ferroptosis [64]. Additionally, we verified that the changes in Fe^2+^ are positively correlated with the expression of TF and TFR. A high lipid diet induces metabolic disturbances in the hepatic tissue of largemouth bass, resulting in increased iron deposition. Ferroptosis may be one of the mechanisms underlying the pathogenesis of hepatic steatosis.

Soybean oil is a widely used lipid source in aquafeed due to its high levels of n-3 PUFAs, which are crucial to fish growth and reproduction [65]. However, excessive intake of polyunsaturated fatty acids can affect fish growth, lipid metabolism, and liver antioxidant capacity [66]. PUFAs are highly sensitive to lipid peroxidation [67], and various fatty acids may engage in the ferroptosis pathway [68]. Therefore, the content of PUFAs within cells may determine the status of lipid peroxidation, thereby determining the extent of ferroptosis. In studies involving the spotted knifejaw [13] and Atlantic cod *Gadus morhua* [69], the levels of LC-PUFAs in the liver and muscles increased significantly with rising dietary lipid levels. However, elevated dietary lipid levels had adverse effects on the deposition and synthesis of LC-PUFAs in fish. A gene screening identified two lipid metabolism regulatory factors, LPCAT3 and ACSL4, playing significant roles in driving ferroptosis processes [70]. ACSL4 is a member of the ACSL family, and PUFAs can be combined into PUFA-CoA under its action. Then, through LPCAT3 catalysis, these are transformed into esterified compounds (PUFA-PLs), which can subsequently undergo lipid peroxidation under the action of LOX12 [71,72,73,74]. Therefore, ACSL4 and LPCAT3 play crucial roles in the occurrence of iron death. This process reduces the production of free fatty acids, thereby decreasing lipid peroxidation and promoting intracellular lipid metabolism reactions, leading to a reduction in intracellular lipid accumulation and the protection of cell membranes from damage caused by oxidative reactions and iron death. Inhibition of ACSL4 and LPCAT3 expression may be a primary mechanism for reducing cellular sensitivity to iron death, regulated by multiple signaling pathways [21,75]. Our investigation uncovered a correlation between elevated dietary lipid levels and an upward trend in the mRNA expression and protein levels of ACSL4, LPCAT3, and LOX12. It has been confirmed that the ACSL4/LPCAT3/LOX12 axis is potentially implicated in the mechanism underlying ferroptosis triggered by a diet rich in lipids; the precise mechanism underlying this involvement necessitates further investigation. Comparable research has been documented [76], as transcriptomic analysis of fatty liver disease in gibel carp reveals the potential activation of ferroptosis [19].

## 5. Conclusions

The current study (crude protein 47%, initial body weight 14.36 g) indicated that a dietary lipid content of 12.2% is ideal for maximizing largemouth bass growth performance. Excessive dietary lipid levels up-regulate the ACSL4/LPCAT3/LOX12 axis and promote oxidative stress and ferroptosis in the liver as well as decrease growth performance in fish. However, due to limitations in the research design, the potential mechanism of the axis on fish remains to be further investigated (Figure 6).

## Figures and Tables

**Figure 1 antioxidants-13-00808-f001:**
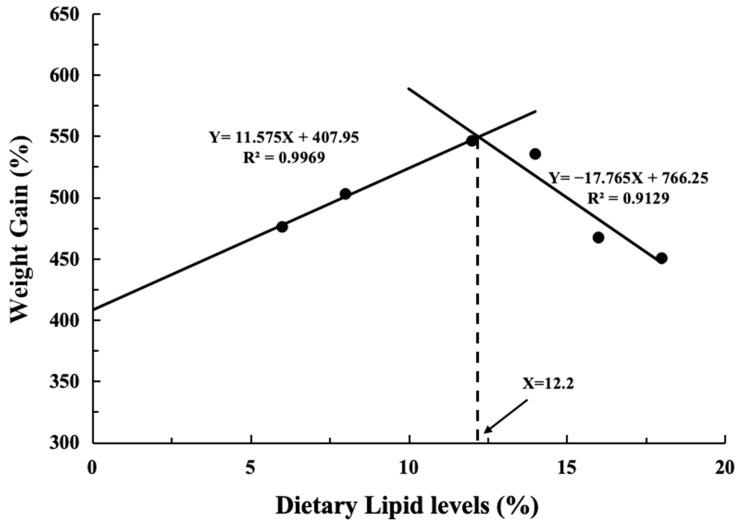
The largemouth bass given the experiment diets showed a relationship between dietary lipid levels and percent weight increase (WG, %).

**Figure 2 antioxidants-13-00808-f002:**
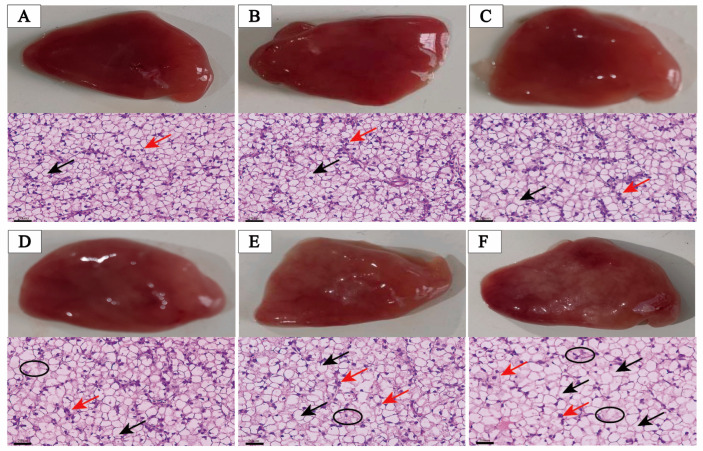
Anatomy and histological analysis of livers of largemouth bass fed different lipid level diets ((**A**): 6.45%; (**B**): 8.58%; (**C**): 12.46%; (**D**): 14.56%; (**E**): 16.37%; (**F**): 18.24%). Liver morphology and paraffin-embedding prepared liver sections, stained with hematoxylin and eosin (black arrow: fat drop; red arrow: nuclear migration; and black circle: cytolysis). Scale bars = 20 µm (400×).

**Figure 3 antioxidants-13-00808-f003:**
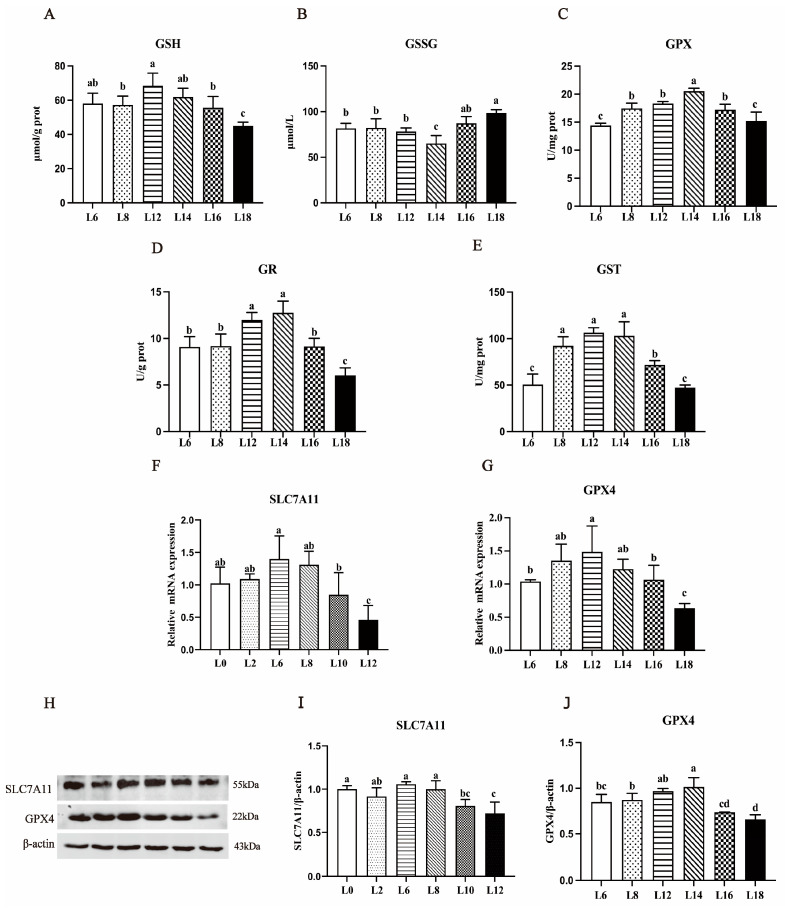
Effect of varying dietary lipid composition on largemouth bass liver antioxidant capability. The content of GSH (**A**) and GSSG (**B**) in the livers of largemouth bass. Largemouth bass liver enzyme activity measured by glutathione peroxidase (GPX, (**C**)), glutathione reductase (GR, (**D**)), and glutathione S-transferase (GST, (**E**)). Solute carrier family 7 member 11 (SLC7A11) and GPX4 (glutathione peroxidase 4) mRNA expression and protein levels in largemouth bass liver tissue (**F**–**J**). The data show three replicates’ means ± SD. Bars labeled with unique letters represent statistically significant differences at a significance level of *p* < 0.05.

**Figure 4 antioxidants-13-00808-f004:**
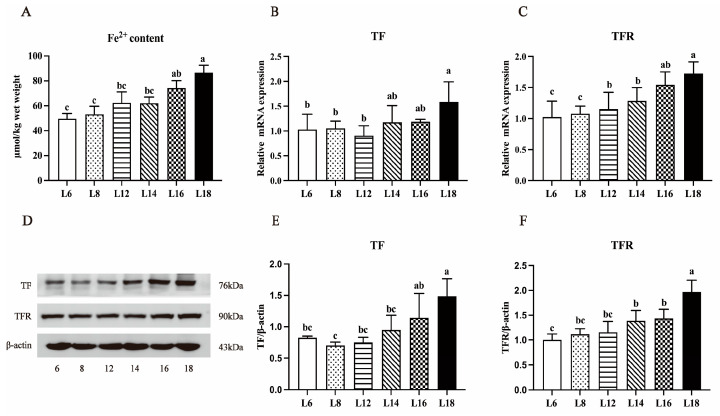
Effects of different lipid levels on liver iron homeostasis of largemouth bass. The content of Fe^2+^ in liver tissues (**A**). The mRNA expression (**B**,**C**) and protein levels (**D**–**F**) of TF (Transferrin) and TFR (Transferrin receptor) in liver tissues. Data represent means ± SD of three replicates. Bars with different letters indicate significant differences (*p* < 0.05).

**Figure 5 antioxidants-13-00808-f005:**
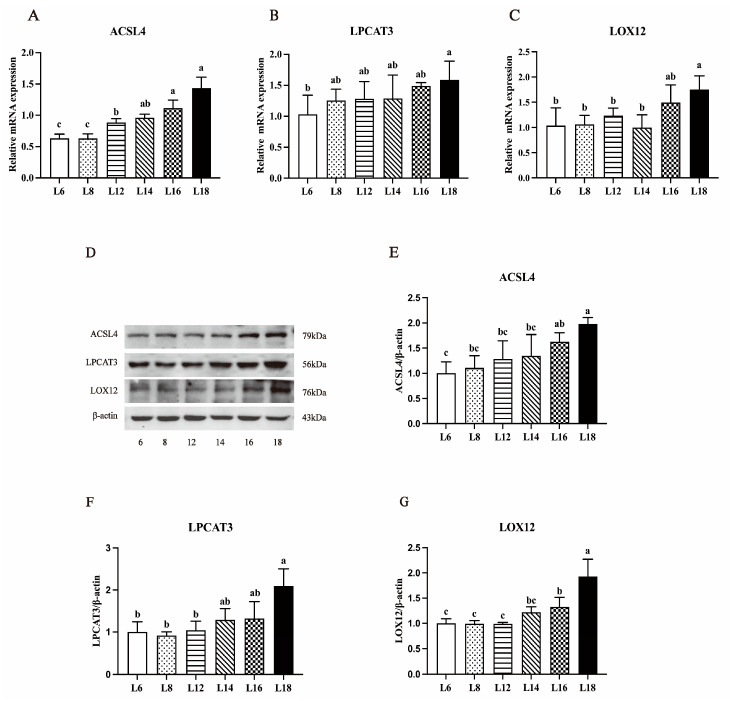
The mRNA expressions of ACSL4 (Acyl-CoA synthetase long-chain family member 4, (**A**)), LPCAT3 (Lysophosphatidylcholine acyltransferase 3, (**B**)), and LOX12 (Lipoxygenase-12, (**C**)) in the livers of largemouth bass with dietary different lipid levels. Western blotting images of ACSL4, LPCAT3, LOX12 and β-antin (**D**). The protein levels of ACSL4 (**E**), LPCAT3 (**F**), and LOX12 (**G**) were analyzed by Western blotting in the livers of largemouth bass with dietary different lipid levels. Values are expressed as the mean ± SD of three replicates. Bars labeled with different letters represent statistically significant differences (*p* < 0.05).

**Figure 6 antioxidants-13-00808-f006:**
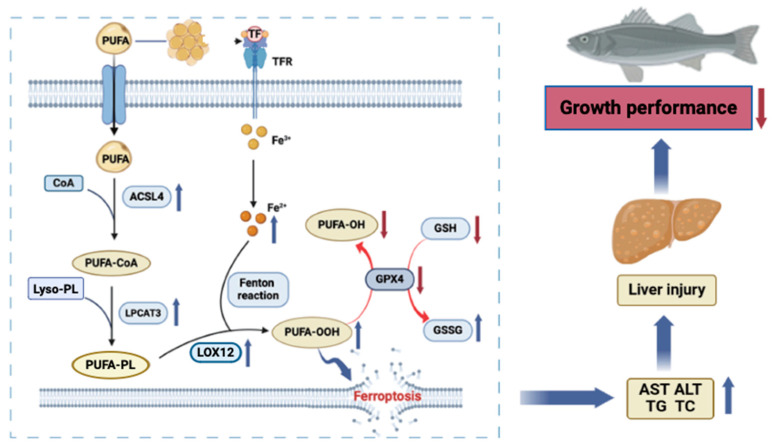
Mechanisms by which excess lipid levels promote hepatic ferroptosis in largemouth bass. PUFA (polyunsaturated fatty acids), ACSL4 (Acyl-CoA synthetase long-chain family member 4), PUFA-CoA (polyunsaturated fatty acids-Coenzyme A), lyso-PL (lysophospholipid), LPCAT3 (Lysophosphatidylcholine acyltransferase 3), PUFA-PL (polyunsaturated fatty acids-phospholipid), LOX12 (Lipoxygenase 12), GPX4 (glutathione peroxidase), PUFA-OH (lipid alcohols), PUFA-OOH (lipid hydroperoxide), GSSG (oxidized glutathione), GSH (reduced glutathione), ALT (Alanine aminotransferase), AST (Aspartate transaminase), TG (triglyceride), and TC (total cholesterol).

**Table 1 antioxidants-13-00808-t001:** Dietary composition and nutritional content.

Ingredients (g kg^−1^) ^1^	Dietary Lipid Levels (%)					
	6	8	12	14	16	18
Soybean oil	0	20	60	80	100	120
Microcrystalline cellulose	127	107	67	47	27	7
Fishmeal	430	430	430	430	430	430
Chicken powder	70	70	70	70	70	70
Pork powder	90	90	90	90	90	90
Gluten	70	70	70	70	70	70
Fermented soybean meal	50	50	50	50	50	50
Wheat meal	50	50	50	50	50	50
Cassava starch	50	50	50	50	50	50
Lysine	2	2	2	2	2	2
DL-Methionine	2	2	2	2	2	2
CaH_2_PO_4_	10	10	10	10	10	10
Phospholipid oil	20	20	20	20	20	20
Choline chloride	4	4	4	4	4	4
Vitamin premix ^2^	15	15	15	15	15	15
Mineral premix ^3^	10	10	10	10	10	10
Nutrients content (%) ^4^						
Crude protein	47.15	47.22	47.25	47.24	47.64	47.50
Crude lipid	6.45	8.58	12.46	14.56	16.37	18.24
Ash	16.49	16.37	16.43	16.90	16.24	16.55
Moisture	7.46	7.31	7.39	7.55	7.46	7.36
Gross energy (MJ kg^−1^)	16.05	16.89	18.41	19.23	20.02	20.72

^1^ Soybean oil (Baker Commodities Co., Ltd., Chengdu, China), fishmeal (Chengdu Meiyide Biotechnology Co., Ltd., Chengdu, China), chicken powder (Sichuan Xinrui Feed Technology Co., Ltd., Chengdu, China), pork powder, gluten (Dongguan Yihai Kerry Serui Starch Co., Ltd., Dongguan, China), fermented soybean meal (Shandong Hengfeng Lebang Biological Technology Co., Ltd., Linyi, China), wheat meal (Sichuan Wanfeng Cereals and Oils Co., Ltd., Deyang, China), cassava starch (Xionghe Beiguang Co., Ltd., Chengdu, China), Lysine, DL-Methionine (Chuanheng Ecological Technology Co., Ltd., Deyang, China), CaH_2_PO_4_ (Chuanheng Ecological Technology Co., Ltd., Deyang, China). Fish meal (66.94% crude protein, 8.15% crude lipid); chicken powder (59.67% crude protein, 12.31% crude lipid); pork powder (64.18% crude protein, 20.81% crude lipid); gluten (78.7% crude protein, 1.33% crude lipid); fermented soybean meal (45.67% crude protein, 3.46% crude lipid); and wheat meal (13.46% crude protein, 1.61% crude lipid). ^2^ Vitamin premix (g kg^−1^ premix): vitamin A, 8.89 g; vitamin E, 14.67 g; vitamin B12, 6.67 g; vitamin D, 53.33 g; Folic acid, 0.13 g; Biotin, 3.33 g; Inositol, 27.21 g; vitamin B1, 1.40 g; Riboflavin, 2.92 g; Pyridoxine, 1.47 g; vitamin K3, 1.39 g; Pantothenic acid, 3.89 g; niacin, 3.00 g; and vitamin C, 110.53 g. All ingredients were diluted with cassava starch to 1.00 kg. ^3^ Mineral premix (g kg^−1^ premix): FeSO_4_·H_2_O, 10.00 g; MgSO_4_·H_2_O, 52.82 g; CuSO_4_·5H_2_O, 2.00 g; ZnSO_4_·H_2_O, 11.59 g; MnSO_4_·H_2_O, 3.77 g; CoCl_2_·6H_2_O, 8.06 g; Ca(IO_3_)_2_, 6.13 g; Na_2_SeO_3_, 44.44 g; KH_2_PO_4_, 17.40 g; and NaCl, 7.63 g. All ingredients were diluted with CaCO_3_ to 1.00 kg. ^4^ Crude protein, crude lipid, and ash were measured using the Association of Official Analytical Chemists (AOAC) methods.

**Table 2 antioxidants-13-00808-t002:** Primer sequences of genes chosen for Real-time PCR analysis.

Name ^1^	Sequences	Tm (°C) ^2^	Accession Number	E (%) ^3^
ACSL4-QF	GATCTGCACTCACCCCGACA	61.4	XM_038699899	92.3
ACSL4-QR	GCTCTGGACTCAAATGCACCT			
LPCAT3-QF	CAGCCCTTCTGGTATCGTTG	63.3	XM_038711111	99.3
LPCAT3-QR	ATACACCCTCCGCTATAACCC			
LOX12-QF	ATGGTGCATACCTGACACCTC	61.4	XM_038731340	92.6
LOX12-QR	TCCCTCACTTGGCCTTTCTTG			
TF-QF TF-QR	GGGCAACAATCCCCAAACT TCATCCACCAGACACTGAAAGG	61.4	XM_038718037	94.4
TFR-QF TFR-QR	CTTCCTGTCGCCCTATGAGTC GTCTGCCTTAGGGTTGTTGGT	64.5	XM_038718573	100.4
GPX4-QF	GTTTACGCATCCTTGCCTTCC	59.0	XM_038716292	97.7
GPX4-QR	GCTCTTTCAGCCACTTCCACAA			
SLC7A11-QF SLC7A11-QR	GGGGCTACAGATCACACGAG ACACTACAGCCCCTTTGACC	57	XM_038699722	95.8
β-actin-QF	CCCCATCCACCATGAAGA	55.7	XM_038695351	93.6
β-actin-QR	CCTGCTTGCTGATCCACAT			
18S-QF	TGAATACCGCAGCTAGGAATAATG	59.0	MH_018569.1	99.5
18S-QR	CCTCCGACTTTCGTTCTTGATT			

^1^ ACSL4 (Acyl-CoA synthetase long-chain family member 4), LPCAT3 (Lysophosphatidylcholine acyltransferase 3), LOX12 (Lipoxygenase-12), TF (Transferrin), TFR (Transferrin receptor), GPX4 (glutathione peroxidase 4) and SLC7A11(Solute carrier family 7 member 11); ^2^ Tm (melting temperature, °C); and ^3^ E (amplification efficiency, %).

**Table 3 antioxidants-13-00808-t003:** IBW, FBW, PWG, SGR, FI, FE, FCR, PER, CF, VSI, and LSI of largemouth bass fed diets with graded levels of lipid (g kg^−1^ diets) for 11 weeks ^1^.

Items	Dietary Lipid Levels					
	6	8	12	14	16	18
IBW ^2^	14.39 ± 0.13	14.31 ± 0.07	14.38 ± 0.03	14.37 ± 0.02	14.31 ± 0.16	14.40 ± 0.07
FBW	82.86 ± 1.00 ^bc^	86.28 ± 3.80 ^b^	92.93 ± 3.05 ^a^	91.33 ± 0.42 ^a^	81.16 ± 1.11 ^bc^	79.24 ± 2.07 ^c^
SR	98.33 ± 1.70	98.67 ± 1.25	99.67 ± 0.47	99.33 ± 0.47	99.33 ± 0.47	99.00 ± 1.41
WG	475.91 ± 10.55 ^cd^	502.87 ± 27.71 ^bc^	546.10 ± 19.73 ^a^	535.48 ± 3.50 ^ab^	467.28 ± 3.42 ^cd^	450.45 ± 14.78 ^d^
SGR	2.27 ± 0.02 ^cd^	2.33 ± 0.06 ^bc^	2.42 ± 0.04 ^a^	2.40 ± 0.01 ^ab^	2.25 ± 0.01 ^d^	2.21 ± 0.03 ^d^
FI	61.54 ± 2.13 ^bc^	63.76 ± 5.18 ^ab^	67.55 ± 2.21 ^a^	67.20 ± 1.31 ^a^	59.53 ± 0.42 ^bc^	58.55 ± 2.17 ^c^
FE	111.31.24 ± 1.85 ^b^	112.98 ± 1.93 ^ab^	116.27 ± 2.71 ^a^	114.53 ± 1.49 ^ab^	112.30 ± 1.83 ^ab^	110.78 ± 2.98 ^b^
FCR	89.85 ± 1.51 ^a^	88.53 ± 1.50 ^ab^	86.04 ± 2.02 ^b^	87.32 ± 1.14 ^ab^	89.06 ± 1.44 ^ab^	90.31 ± 2.46 ^a^
PER	2.40 ± 0.04	2.37 ± 0.01	2.33 ± 0.03	2.34 ± 0.04	2.36 ± 0.04	2.36 ± 0.02
CF	2.20 ± 0.04	2.17 ± 0.03	2.19 ± 0.06	2.18 ± 0.03	2.20 ± 0.03	2.14 ± 0.06
VSI	7.92 ± 0.13 ^d^	8.37 ± 0.12 ^c^	8.74 ± 0.17 ^bc^	8.91 ± 0.20 ^b^	9.42 ± 0.09 ^a^	9.74 ± 0.29 ^a^
HSI	2.82 ± 0.04 ^b^	2.68 ± 0.07 ^b^	2.99 ± 0.05 ^b^	2.93 ± 0.42 ^b^	3.45 ± 0.13 ^a^	3.64 ± 0.22 ^a^

^1^ Values are the mean ± SD, n = 3. Mean values with different superscripts in the same row are significantly different (*p* < 0.05). ^2^ IBW (initial body weight, g fish^−1^), FBW (final body weight, g fish^−1^), SR (survival rate, %), WG (weight gain, %), SGR (specific growth rate, %/d), FI (feed intake, g fish-1), FE (feed efficiency, %), FCR (feed conversion ratio, %), PER (protein efficiency ratio, %), CF (condition factor, %), VSI (visceral body index, %), HSI (hepatosomatic index, %), VW (viscera weight, g fish^−1^), and LW (liver weight, g fish^−1^). SR (%) = 100 × final fish number/initial fish number; PWG (%) = 100 × (FBW − IBW)/IBW; SGR (%) = 100 × (ln FBW − ln IBW)/days; FI (g) = total feed intake/fish; FE (%) = 100 × (FBW − IBW)/FI; FCR (%) = 100 × FI/(FBW − IBW); PER = (FBW − IBW)/protein intake (g); CF (%) = 100 × FBW/body length^3^(cm); VSI (%) = 100 × VW/FBW; and HSI (%) = 100 × LW/FBW.

**Table 4 antioxidants-13-00808-t004:** Whole body composition (%, dry-basis) of largemouth bass fed diets with different levels of lipids (%) for 11 weeks ^1^.

Items	Dietary Lipid Levels					
	6	8	12	14	16	18
Moisture	70.76 ± 0.52	70.61 ± 0.24	70.69 ± 0.19	70.26 ± 0.37	70.92 ± 0.46	70.93 ± 0.24
Protein	17.28 ± 0.37 ^d^	18.41 ± 0.15 ^b^	19.42 ± 0.33 ^a^	18.93 ± 0.66 ^a^	18.00 ± 0.48 ^bc^	17.85 ± 0.13 ^c^
Lipid	6.88 ± 0.41 ^d^	7.56 ± 0.57 ^c^	8.17 ± 0.20 ^b^	8.65 ± 0.29 ^ab^	8.57 ± 0.48 ^ab^	8.94 ± 0.19 ^a^
Ash	4.19 ± 0.12 ^a^	4.00 ± 0.04 ^b^	3.67 ± 0.06 ^cd^	3.72 ± 0.10 ^c^	3.59 ± 0.11 ^d^	3.57 ± 0.05 ^d^

^1^ The values are the means ± SD of three replicate groups, each containing six fish. Significant differences exist between values with different superscripts in the same rows (*p* < 0.05).

**Table 5 antioxidants-13-00808-t005:** Plasma AST and ALT activity, liver TG, and TC content of fish fed diets with graded levels of lipids (%) for 11 weeks ^1^.

Items	Dietary Lipid Levels					
	6	8	12	14	16	18
Plasma ^2^						
ALT	10.32 ±2.57 ^b^	12.14 ± 3.97 ^b^	14.19 ± 2.68 ^b^	14.20 ± 1.52 ^b^	20.11 ± 3.45 ^a^	21.34 ± 0.51 ^a^
AST	33.53 ± 1.88 ^c^	37.14 ± 2.26 ^bc^	37.58 ± 1.74 ^bc^	38.87 ± 1.10 ^b^	41.51 ± 3.44 ^ab^	43.97 ± 2.49 ^a^
Liver						
TG	0.157 ± 0.012 ^b^	0.158 ± 0.008 ^b^	0.191 ± 0.005 ^b^	0.192 ± 0.032 ^b^	0.256 ± 0.030 ^a^	0.292 ± 0.013 ^a^
TC	0.020 ± 0.006 ^d^	0.028 ± 0.008 ^cd^	0.040 ± 0.014 ^bc^	0.054 ± 0.010 ^ab^	0.055 ± 0.008 ^ab^	0.070 ± 0.006 ^a^

^1^ Values are means ± SD. Significant differences existed between values in the same row with various superscripts (*p* < 0.05; n = 3). ^2^ ALT (Alanine aminotransferase, U/L), AST (Aspartate transaminase, U/L), TG (triglyceride, mmol/g prot), and TC (total cholesterol, mmol/g prot).

**Table 6 antioxidants-13-00808-t006:** MDA and PC content, T-SOD, CAT, ASA, and AHR activities in the livers of fish fed diets with graded levels of lipid (%) for 11 weeks ^1^.

Items	Dietary Lipid Levels					
	6	8	12	14	16	18
MDA ^2^	0.95 ± 0.02 ^b^	0.75 ± 0.08 ^c^	0.76 ± 0.09 ^c^	0.86 ± 0.06 ^bc^	1.27 ± 0.09 ^a^	1.28 ± 0.14 ^a^
PC	7.50 ± 1.20 ^ab^	6.44 ± 0.70 ^b^	4.90 ± 0.86 ^c^	4.75 ± 0.51 ^c^	7.55 ± 0.47 ^ab^	8.23 ± 0.81 ^a^
T-SOD	160.99 ± 11.87 ^abc^	166.64 ± 5.93 ^ab^	165.96 ± 5.83 ^ab^	172.76 ± 6.26 ^a^	154.29 ± 11.03 ^bc^	147.04 ± 7.27 ^c^
CAT	9.85 ± 0.33 ^c^	11.33 ± 0.21 ^b^	12.35 ± 0.43 ^a^	12.07 ± 0.43 ^a^	11.25 ± 0.28 ^b^	10.86 ± 0.11 ^b^
ASA	433.08 ± 13.78 ^d^	441.95 ± 12.53 ^d^	728.31 ± 19.85 ^b^	824.39 ± 73.38 ^a^	730.97 ± 15.50 ^b^	627.67 ± 11.02 ^c^
AHR	122.04 ± 4.50 ^d^	130.49 ± 5.34 ^cd^	141.91 ± 2.52 ^b^	167.96 ± 6.45 ^a^	134.15 ± 4.82 ^bc^	93.58 ± 1.93 ^e^

^1^ Values are means ± SD. Values within the same row bearing different superscripts indicate significant differences (*p* < 0.05; n = 3). ^2^ MDA (malondialdehyde, mmol/mg prot), PC (protein carbonyl, mmol/mg prot), T-SOD (total superoxide dismutase, U/mg prot), CAT (catalase, U/mg prot), ASA (anti-superoxide anion, ug/mg prot), and AHR (anti-hydroxyl radical, U/mg prot).

## Data Availability

The data presented in this study are available in this manuscript.

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
