# Peer review of "Hepatic Oxidative Stress and Cell Death Influenced by Dietary Lipid Levels in a Fresh Teleost"

_antioxidants, 2024, doi:10.3390/antiox13070808_

Round 1

Reviewer 1 Report

This study aims to assess the potential impact of varying lipid concentrations on growth outcomes, liver health effects, and potential mechanisms in largemouth bass. The findings from this study hold significant implications for formulating efficient feed and enhancing the health of cultured largemouth bass. Overall, the MS is well written, however, there are still some shortcomings in the manuscript, which should be carefully revised before publication.

There is a problem with the author column, which needs to be checked carefully. Jun Jiang 1, 3 and *?

based on weight gain, line regression analysis established the optimal lipid level at 12.2%

Table 1. Add the gross energy of feed.

Table 1. Provide nutrient levels and sources of major raw materials

6%, 8%, 12%, 14%, 16%, and 18%. What are the design principles for this fat level gradient? 8% to 12% why is it different from other groups?

Why the lipid source used in this experiment is soybean oil, why not choose fish oil, or the combination of fish oil and soybean oil, most of the current feed formula is still practical fish oil.

Meticulously sifted through a sieve with a mesh size of 60? Generally speaking, sifted through a sieve with a mesh size of 80. Fineness of grinding is particularly important for digestion and utilization of fish.

Add the amplification efficiency of each gene in table 2.

Why choose broken-line analysis. Add clarification in the data analysis.

Add the results of FCR in table 3.

Add the units of the indicator in table 5.

Add the units of the indicator in table 6.

Check the unit of liver antioxidant index in Figure 3.

Line 334. Should be [25].?

Author Response

We thank the reviewer for the time in closely viewing our manuscript, and for the detailed constructive comments. All of the suggestions have been adopted in the revision.

Reviewer 2 Report

  1. Also, based on the GSH content in Figure 3, iron accumulation in Figure 4, and MDA content in Table 6, all of the alterations by lipid inclusion are not dramatic despite significance in statistics by the authors. It appears that an increase lipid level in diet rises the lipid availability for the susceptibility of pro-oxidation by ROS, while not really induces overt ferroptotic death.    
  2.  
  3.  
  1. A morphological study by Prussian blue staining for iron accumulation is required to confirm iron accumulation within cells or in the interrestrial space. An iron content increase by biochemical analysis cannot account for whether iron is accumulated due to heme or RBC accumulation in the central vein or capillary.
  2. In Figure 2 panel B, enlarged image should be shown to note the significant differences.
  3. In all figures and tables, the results were expressed as means ± SEM. Lots of values with SE or error bar overlaps, and were claimed for significant differences. The statistics should be rechecked.
  4. The authors need to elucidate how increased lipid levels in diet promote iron accumulation and TF/TFR pathway leading to iron accumulation a pro-oxidant to trigger ferroptotic program?
  5. Also, the authors need to validate iron as the source to trigger oxidative stress in the liver by treating fish with iron chelators such deferoxamine or dexrazoxane, or lipophilic antioxidants ferrostatin-1 or liproxstatin-1.
  6. Also, based on the GSH content in Figure 3, iron accumulation in Figure 4, and MDA content in Table 6, all of the alterations by lipid inclusion are not dramatic despite significance in statistics by the authors. It appears that an increase lipid level in diet rises the lipid availability for the susceptibility of pro-oxidation by ROS, while not really induce overt ferroptotic death.     
  7. The hallmarks of ferroptosis is excessive lipid peroxidation, iron accumulation oxidative stress. The authors need to show the alternative iron source such as Heme/HO-1 independent of IF/IFR pathway. Also SLC7A11/xCT should be examined to elucidate suppressed GSH levels.
  8.  

Author Response

Thank you very much for your helpful comments regarding the weak aspects, which has afforded us the opportunity to refine and enhance our manuscript. Please review attached the reply.

Reviewer 3 Report

The authors examined the impact of different dietary lipid levels on the growth performance, body composition, and liver damage as well oxidative stress and ferroptosis biomarkers in M. salmoides.

The authors found that a dietary lipid content of 12.2% is ideal for maximizing the fish growth performance and that higher dietary lipid levels induce oxidative stress as well as alter the levels of ACSL4, LPCAT3, LOX12 and hepatic ferroptosis, liver damage and fish growth performance.

The study is well-designed and conducted and the paper is very-well written, but the performed experiments do to allow establishing cause and consequence relationship and conclusion on the role of ACSL4, LPCAT3, LOX12 axis in hepatic ferroptosis, liver damage and fish growth performance induced by dietary fat. To address the role of this axis intervention experiments such as blocking ACSL4, LPCAT3, LOX12 activities should be performed.

If this is not possible the authors should change the conclusion and add a limitations section saying that due to study design the present results preclude mechanistic approach to substantiate the role of the ACSL4, LPCAT3, LOX12 axis.

The study is well-designed and conducted and the paper is very-well written, but the performed experiments do to allow establishing cause and consequence relationship and conclusion on the role of ACSL4, LPCAT3, LOX12 axis in hepatic ferroptosis, liver damage and fish growth performance induced by dietary fat. To address the role of this axis intervention experiments such as blocking ACSL4, LPCAT3, LOX12 activities should be performed.

If this is not possible the authors should change the conclusion and add a limitations section saying that due to study design the present results preclude mechanistic approach to substantiate the role of the ACSL4, LPCAT3, LOX12 axis.

There are, however some weaknesses in data presentation that need improvement.

Table 1:

What are the units for the data presented, such as Soybean oil and the others in the column.

Tables 2 and 3:

Please list and explain all abbreviations used in the table in the table footer.

Lane 230:

‘’The biochemical parameters of largemouth bass PLASMA AND liver…..’’

Table 5:

Please add units for the measured parameters as well as list and explain all abbreviations used in the table in the table footer.

Figure 2:

….localized whitening as well as fat droplets, nuclear migration and cytolysis are not visible in the presented images. Better and larger images are needed for Fig. 2.

Legend Fig. 4, 5, and 6:

Please explain abbreviations.

Lane 449: ….by binding PUFA-CoA; this statement is not correct as the role of ASCL4 is, as stated in the Introduction section, the formation of acyl-CoA i.e. PUFA-CoA.

Fig. 7:

Please explain abbreviations in the legend.

In the scheme the authors indicated LPCAT3-mediated generation of PUFA-PE.

Since LPCAT3 catalyzes through acylation of LPC and other lysophospholipids (lyso-PL) the formation of primarily, but not exclusively PUFA-PC, the authors should replace in the scheme PE with lyso-PL as well as PUFA-PE with PUFA-PL.   

Author Response

Thank you very much for taking the time to review our manuscript.  We have carefully modify them according to your comments. Thank you again for your constructive comments on our manuscript.

Round 2

Reviewer 2 Report

  1. Since the authors also agreed the induction by dietary lipid levels in not enough to result in overt ferroptosis. The title is suggested as “Hepatic oxidative stress and cell death influenced by dietary lipid levels in a fresh teleost” Also, in line 28 of the abstract, “induce hepatic oxidative stress and promote cell death through a ferroptotic-like program,…”

2. Most of the comments are answered

  1. Since the authors also agreed the induction by dietary lipid levels in not enough to result in overt ferroptosis. The title is suggested as “Hepatic oxidative stress and cell death influenced by dietary lipid levels in a fresh teleost” Also, in line 28 of the abstract, “induce hepatic oxidative stress and promote cell death through a ferroptotic-like program,…”

2. Most of the comments are answered

Author Response

Thank you very much for reviewing our manuscript again and for providing us with valuable feedback and suggestions. We attach great importance to your opinions and have made corresponding revisions and improvements to the manuscript based on your guidance.

Reviewer 3 Report

The authors improved the manuscript, but there are still some issues that need to be addressed.

The authors improved the manuscript, but there are still some issues that need to be addressed.

Abstract:

There are many unexplained abbreviatins; they need to be explained or, some of them may be replaced with a term biomarkers of oxidatiove stress.

Lanes: 26-29: The sentence is too long, gramatically not correct and not clear; please re-writte it to be simple and informative.

it is linear regression not line regression

Revised Figure 6 in the revised manuscript: There are still PUFA-PE and PE in the scheme. In the repsonse letter in that scheme PUFA-PE is correctly replaced with PUFA-PL but PE is still not replaced with lyso-PL

In Summary regarding Fig. 6: Please replace PE with lyso-PL and PUFA-PE with PUFA-PL

Author Response

We are truly grateful for your second review of our manuscript and the invaluable feedback and insightful suggestions you have kindly offered.We acknowledge and accept all your comments and suggestions and have revised them in the revised manuscript.
